# Improved state-level influenza nowcasting in the United States leveraging Internet-based data and network approaches

Fred S. Lu [1], Mohammad W. Hattab[2], Cesar Leonardo Clemente[3], Matthew Biggerstaff[4] & Mauricio Santillana[1,5]

In the presence of health threats, precision public health approaches aim to provide targeted, timely, and population-specific interventions. Accurate surveillance methodologies that can estimate infectious disease activity ahead of official healthcare-based reports, at relevant spatial resolutions, are important for achieving this goal. Here we introduce a methodological framework which dynamically combines two distinct influenza tracking techniques, using an ensemble machine learning approach, to achieve improved state-level influenza activity estimates in the United States. The two predictive techniques behind the ensemble utilize (1) a self-correcting statistical method combining influenza-related Google search frequencies, information from electronic health records, and historical flu trends within each state, and (2) a network-based approach leveraging spatio-temporal synchronicities observed in historical influenza activity across states. The ensemble considerably outperforms each component method in addition to previously proposed state-specific methods for influenza tracking, with higher correlations and lower prediction errors.

[1] Computational Health Informatics Program, Boston Children's Hospital, Boston, MA 02115, USA. [2] Wyss Institute for Biologically Inspired Engineering, Harvard Medical School, Boston, MA 02115, USA. [3] Tecnológico de Monterrey, 64849 Monterrey, N.L., Mexico. [4] Influenza Division, National Center for Immunization and Respiratory Disease, Centers for Disease Control and Prevention, Atlanta, GA 30333, USA. [5] Department of Pediatrics, Harvard Medical School, Boston, MA 02115, USA. Correspondence and requests for materials should be addressed to F.S.L. (email: fredlu.flac@gmail.com) or to M.S. (email: msantill@g.harvard.edu)

The Internet has enabled near-real time access to multiple sources of medically relevant information, from cloud-based electronic health records to environmental conditions, social media activity, and human mobility patterns. These data streams, combined with an increase in computational power and our ability to process and analyze them, promise to revolutionize the identification and delivery of community-level interventions in the presence of health threats. As the field of precision medicine[1] continues to yield important medical insights as a consequence of improvements in the quality and cost of genetic sequencing as well as advances in bioinformatics methodologies, precision public health efforts aim to eventually provide the right intervention to the right population at the right time[2]. In this context, real-time disease surveillance systems capable of delivering early signals of disease activity at the local level may give local decision-makers, such as governments, school districts, and hospitals, valuable and timely information to better mitigate the effects of disease outbreaks. Our work focuses on a methodology aimed at achieving this for influenza activity surveillance.

Influenza has a large seasonal burden across the United States, infecting up to 35 million people and causing between 12,000 and 56,000 deaths per year[3]. Limiting the spread of outbreaks and reducing morbidity in those already infected are crucial steps for mitigating the impact of influenza. To guide this effort, public health officials, as well as the general public, should have access to localized, real-time indicators of influenza activity. Established influenza reporting systems currently exist over large geographic scales in the United States, coordinated by the Centers for Disease Control and Prevention (CDC). These systems provide weekly reports of influenza statistics, aggregated over the national, regional (10 groups as defined by the Health and Human Services), and starting in fall 2017, state level. Of particular interest, US Outpatient Influenza-like Illness Surveillance Network (ILI-Net) records the percentage of patients reporting to outpatient clinics with symptoms of influenza-like illness (ILI), which is defined by fever over 100 °F in addition to sore throat or cough, over the total number of patient visits[4]. While these measurements are an established indicator of historical ILI activity, they are frequently revised and require around a week to collect from healthcare providers across the country, analyze, and report. This delay and potential subsequent revisions can reduce the utility of the system for real-time situational awareness and data analysis.

To address this delay, research teams have devised methods to estimate ILI a week ahead of healthcare-based reports and in near-real time, termed nowcasting, at the national and regional levels. These methods incorporate a variety of techniques from statistical modeling and machine learning[5–8], to mechanistic and epidemiological models[8–10]. Many utilize web-based data sources such as Internet search frequencies and electronic health records[5]. Some have also taken into account historical spatial and temporal synchronicities in influenza activity[11,12] to improve the accuracy of existing influenza surveillance tools[13,14]. However, since influenza transmission occurs locally and is spread from person to person, the timing of outbreaks and resulting infection rate curves can significantly differ from state to state. As a result, these successes at nowcasting in national and regional spatial resolutions are likely not enough to aid decision-making at smaller geographic scales, since important information about local conditions is lost in regional or national aggregates.

The first influenza nowcasting system at the state level across the United States was Google Flu Trends (GFT), which operated from 2008 to 2015. GFT reported numerical indicators each week representing influenza activity for each state as well as other geographic resolutions, using Google search activity as a predictor. While innovative at the time, studies have pointed out its large prediction errors when tested in real time and proposed alternative methodologies that can incorporate Google searches more effectively at the national level[6,15–17]. A model replacing GFT for influenza detection, at the state level, was published in 2017 by Kandula, Hsu, and Shaman, who presented retrospective out-of-sample influenza estimates, over the 2005–2011 influenza seasons, using a random forest methodology based on Google searches and historical influenza activity as predictors[18]. While this study showed promise, the authors did not report significant improvements to GFT and provided only aggregate distributional metrics to evaluate the performance of their models over conglomerates of states (as opposed to state-level metrics), making it challenging to replicate or improve their results for any given state. A detailed statistical analysis by Dukic, Lopes, and Polson (2012) presented a Bayesian state-space SEIR model over nine states but did not attempt out-of-sample prediction[19]. Other studies have demonstrated the feasibility of influenza estimation at smaller spatial scales[20–23], though they have not yet extended their results to multiple locations.

In this study, we present a solution for localized influenza nowcasting by first extending to each state a proven methodology for inferring influenza activity, named AutoRegression with General Online information (ARGO), which combines information from influenza-related Google search frequencies, electronic health records, and historical influenza trends[5]. Next, we develop a spatial network approach, named Net, which refines ARGO's influenza estimates by incorporating structural spatio-temporal synchronicities observed historically in influenza activity. Finally, we introduce ARGONet, a novel ensemble approach that combines estimates from ARGO and Net using a dynamic, out-of-sample learning method. We produce retrospective estimates using ARGO from September 2012 to May 2017 and show that ARGO alone demonstrates strong improvement over GFT and an autoregressive benchmark. Then we generate retrospective influenza estimates using ARGONet from September 2014 to May 2017 and show further improvement over ARGO in over 75% of the states studied. We present detailed metrics and figures over each state to enable analysis as well as future refinement of our methods.

## Results

**State-level ARGO models outperform existing benchmarks.** We first adapted the ARGO methodology for state-level influenza detection. ARGO has previously demonstrated the ability to infer influenza activity with high precision over a variety of geographical areas and scales[5,23,24]. The adapted model dynamically fits a regularized multivariable regression on state-level Google search engine frequencies, electronic health record reports from athenahealth, and historical CDC %ILI estimates (see Methods section). We trained a separate ARGO model for each state and used them to generate retrospective out-of-sample estimates from September 30, 2012 to May 14, 2017 for each state in the study.

To assess the predictive performance of ARGO, we produced retrospective estimates for two benchmarks: (a) GFT: the Google Flu Trends time series for each state, fitted to match the scale of each state's CDC %ILI; and (b) AR52: an autoregressive model built on the CDC %ILI using the previous 52 weeks to predict % ILI of the current week (see Methods section). Since autoregression is an important component of ARGO itself, improvement over AR52 indicates the effective contribution of real-time Google search and electronic health record data. Figure 1a compares the performance of ARGO, AR52, and GFT for each state over the time period when GFT estimates were available (September 30, 2012 to August 15, 2015). The three panels display the root mean square error (RMSE), Pearson correlation, and mean absolute percent error (MAPE). ARGO models outperform GFT in RMSE

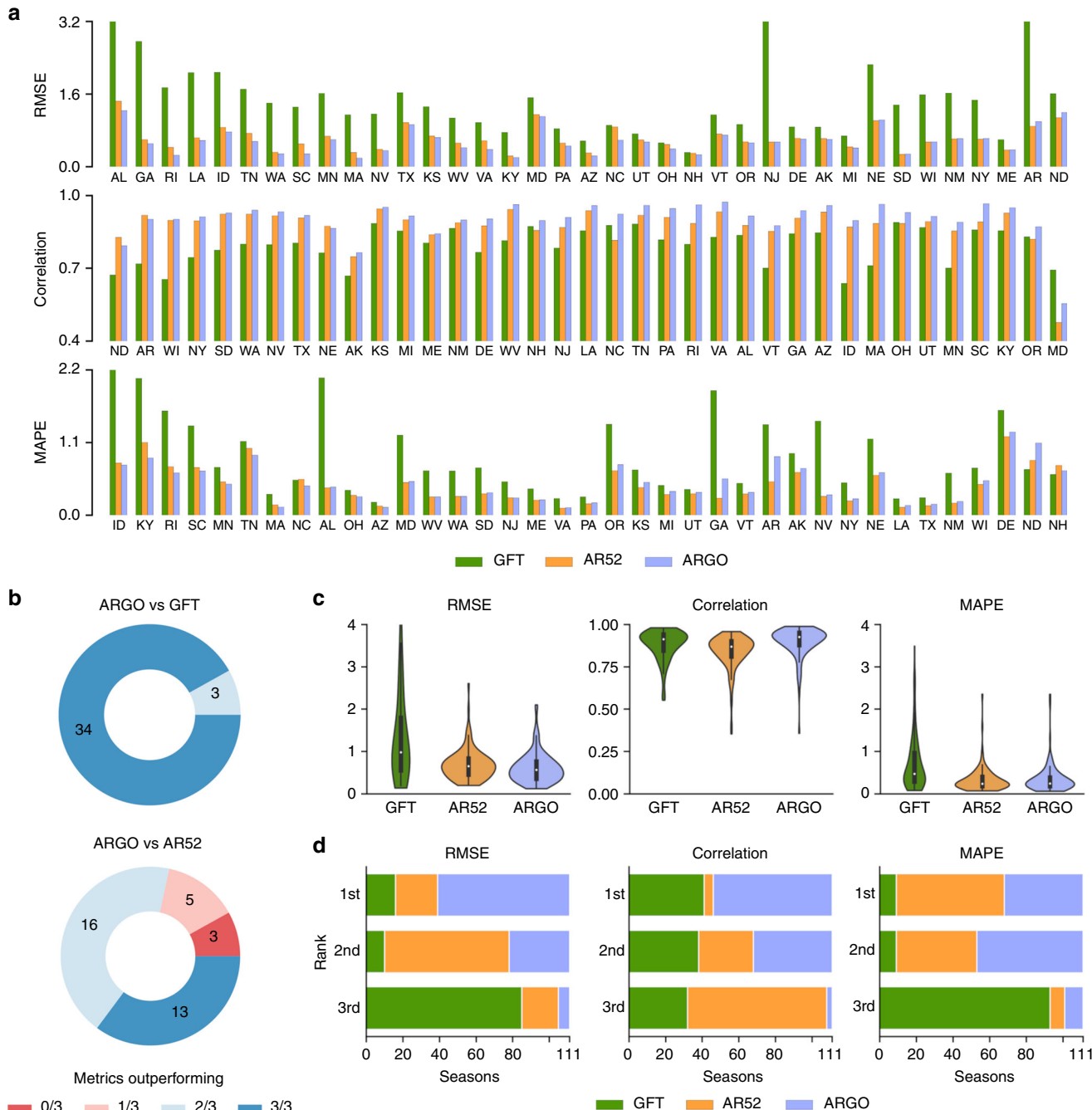

**Fig. 1** Summary of ARGO performance benchmarked with GFT and AR52. **a** State-level performance of ARGO and benchmarks, as measured by RMSE (top), Pearson correlation (middle), and MAPE (bottom), over the period from September 30, 2012 to August 15, 2015. Extreme GFT error values are displayed up to a cutoff point. States are ordered by ARGO performance relative to the benchmarks to facilitate comparison. **b** The proportion of states where ARGO outperforms GFT (top) or AR52 (bottom) in 0, 1, 2, or all 3 metrics. **c** The distribution of values for each metric for each model, over the 111 state-seasons during the same period. The embedded boxplots indicate median and interquartile ranges. Numerical values are reported in Supplementary Table 1. **d** The distribution of ranks attained by each model over the 111 state-seasons for each metric

in every state, in correlation in all but one state, and in MAPE in all but two states. Furthermore, ARGO reduces the RMSE of GFT by >50% in 23 states and increases correlation by >10% in 25 states. ARGO also performs comparably or better in RMSE and correlation than AR52, although it does not generally outperform AR52 in MAPE. In all but eight states, ARGO beats AR52 in a majority (2 or all 3) of metrics (Fig. 1b).

Attention to influenza activity is typically heightened during influenza seasons (between approximately week 40 of one year and week 20 of the next), as the majority of seasonal influenza cases occur within this time frame. We assessed ARGO performance over each influenza season within the same time period, namely the 2012–13 to 2014–15 seasons inclusive. With three seasons where comparison with GFT is available and

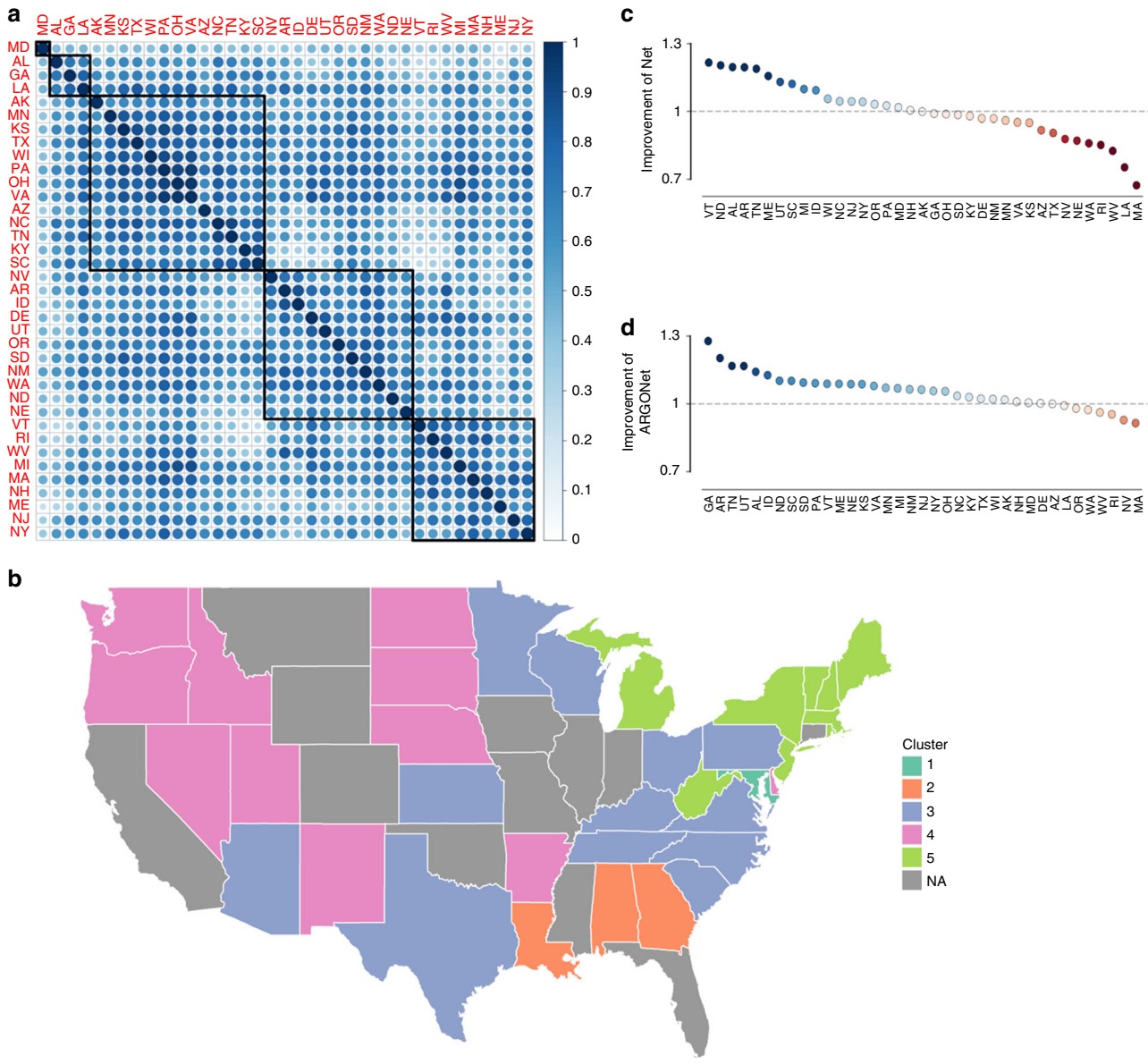

**Fig. 2** Motivation for developing spatio-temporal network and ensemble approaches. **a** Heatmap of pairwise %ILI correlations between all states in the study over the period September 30, 2012 to May 14, 2017. Five clusters of intercorrelated states are denoted by black boxes. **b** Geographic distribution of the five identified clusters. **c** RMSE improvement of Net over ARGO over the period September 28, 2014 to May 14, 2017. The improvement of Net is here defined as the inverse RMSE ratio of Net and ARGO, so values above 1 indicate improvement. **d** RMSE improvement of ARGONet over ARGO over the same period

37 states, this yields 111 state-seasons. Of these, ARGO outperforms GFT in 94 state-seasons in RMSE, 69 in correlation, and 97 in MAPE. ARGO also surpasses AR52 in 83 state-seasons in RMSE, 104 in correlation, and 47 in MAPE (aggregated from Supplementary Table 4). Correspondingly, ARGO outperforms the benchmarks in terms of median and interquartile range over the seasons, with the exception of MAPE against AR52 (Fig. 1c), and ranks first over the majority of state-seasons in RMSE and correlation (Fig. 1d). Interestingly, despite poorer quartile values, GFT has a better tail spread than ARGO in terms of correlation.

**Incorporating spatio-temporal structure in influenza activity.** Because ARGO models the influenza activity within a given state

using only data specific to that state, a natural question is whether information from other states across time can be used to improve the accuracy of influenza predictions. As shown in Fig. 2a, historical CDC %ILI observations show synchronous correlations between states. The clustering of intercorrelated states from the same regions (Fig. 2b) suggests that geographical spatio-temporal structure can be exploited as a correctional effect.

Inspired by this finding, we developed a network-based model on each state, which incorporates multiple weeks of historical % ILI activity from all other states in a regularized multivariable regression. Out-of-sample estimates from this model, denoted Net, improve on the RMSE of ARGO on half of the states over the period of September 28, 2014 to May 14, 2017, but show a comparable increase in error on the other half of the states

(Fig. 2c). Because ARGO and Net dramatically outperform each other over distinct states, we investigated whether an ensemble combining the relative strengths of each model could lead to significant improvement.

**ARGONet ensemble improves on state-level ARGO models.** Our proposed ensemble, denoted ARGONet, dynamically selects either ARGO's or Net's prediction in each week and state based on the past performance of each model over a tuned training space (see Methods section for details). Over the period where ARGONet estimates were generated (September 28, 2014 to May 14, 2017 after a 2-year training window), we found that this approach resulted in out-of-sample improvement in RMSE over ARGO in all but eight states (Fig. 2d). Furthermore, in these eight states, the error increase of ARGONet is relatively controlled compared to the error increase of Net.

In addition to RMSE, ARGONet also displays general improvement in correlation and MAPE over both ARGO and the AR52 benchmark (Fig. 3a). We previously noted that ARGO did not outperform AR52 in MAPE despite being superior in terms of RMSE, which suggests that ARGO is more accurate than AR52 during periods of high influenza incidence and less accurate during low influenza incidence. On the other hand, by incorporating spatio-temporal structure, ARGONet is able to achieve lower MAPE than AR52 over both the entire time period of September 2014–May 2017 and the 108 state-seasons within this period (three states are missing %ILI data for the 2016–17 season, resulting in fewer state-seasons compared to the previous analysis) (Fig. 3a–c). Note that while ARGO and Net outperforms the AR52 benchmark by a majority of metrics in 32 and 30 states, respectively, ARGONet does the same in 36 out of 37 states (Fig. 3b).

Interestingly, the performance increase of ARGONet does not appear to stem from being simultaneously more accurate than ARGO and Net over a majority of state-seasons. Note in Fig. 3d that while ARGONet tends to rank first in a smaller proportion of state-seasons than ARGO or Net, ARGONet ranks either first or second in a far larger proportion of state-seasons than the other two models, indicating that the ensemble's overall success comes from increased consistency. Finally, Fig. 3e subdivides the states by the fraction of seasons (out of 3) where each model outperforms AR52. We see that ARGONet performs favorably (wins 2 or 3 out of 3 seasons) in the vast majority of states, with considerably better distribution in terms of MAPE than ARGO or Net. Refer to Supplementary Table 4 for numerical metrics over each state and season.

Detailed time series comparisons of ARGO and ARGONet relative to the official CDC-reported %ILI values are shown in Fig. 4. Note that our models consistently track the CDC %ILI curve during both high and low periods of ILI activity, whereas GFT often significantly overpredicts during season peaks. Time series plots specifically comparing ARGONet and ARGO over September 2014–May 2017 are presented in Supplementary Figure 1 and better enable the reader to visually inspect ARGONet's improvement over ARGO. In concordance with previous results, ARGONet tracks the CDC %ILI curve more accurately than ARGO over some periods of time, while over other periods the curves are identical. This is an expected result of our winner-takes-all ensemble methodology. The heatmaps under each time series plot in Supplementary Figure 1 indicate which input model was selected by ARGONet in each week.

## Discussion

Our ensemble, ARGONet, successfully combines Google search frequencies and electronic health record data with spatio-temporal trends in influenza activity to produce forecasts with higher correlation and lower errors than all other tested models for current ILI activity at the state level. We believe that the accuracy of our method involves a balance between responsiveness and robustness. Real-time data sources such as Google searches and electronic health records provide information about the present, allowing the model to immediately respond to current influenza trends. On the other hand, using the values of past CDC influenza reports in an autoregression adds robustness by preventing our models from creating outsize errors in prediction. Similarly, incorporating spatial synchronicities adds stability by maintaining state-level inter-correlations evident in historical influenza activity. Our results suggest that dynamic learning ensembles incorporating real-time Internet-based data sources can surpass any individual methods in inferring influenza activity.

Previous work has shown the versatility of ARGO, one of the component models in our ensemble, over a variety of disease estimation scenarios[23,25,26]. At the state level, as shown in this study, it clearly outperformed existing benchmarks over the study period, namely Google Flu Trends and an autoregression. While ARGO alone performs better than the benchmarks, we also found that spatio-temporal synchrony could be used to improve model accuracy (Net). Combining web-based data sources with this structural network-based approach (ARGONet) further improves prediction accuracy and suggests the future study of synchronous network effects at varying geographical scales. Future work may explore adding similar approaches to influenza nowcasting systems at finer spatial resolutions, such as the city level. While previous studies have established the feasibility of city-level flu nowcasting[21–23], the results from this study suggest that ensemble methodologies incorporating flu activity from neighboring cities may improve prediction stability and accuracy at such spatial resolutions as well. This would be of particular interest given that finer spatial resolutions experience fewer numbers of flu cases, which in turn makes the process of finding a meaningful signal in Internet-based data sources a bigger challenge.

A large factor determining the success of our approach is whether covariates that have a strong association with the response variable, over the training set (up to 2 years prior), maintain their behavior in the present. This may be especially important in the network approach, because the selected predictors capture underlying historical geographical relationships. The ILI between two neighboring states may be highly correlated over some seasons, but changes in influenza dynamics over the next season may induce their ILI trends to behave differently. In such cases, the relationship apparent in the older data would provide an inaccurate picture of the present.

Accurate influenza monitoring at the state level faces challenges due to higher variance in data quality across states. The ILINet reporting system within each state varies in reporting coverage and consistency, and thus the magnitudes of influenza activity may not reflect actual differences of influenza activity between states. The quality of Google Trends frequencies and the prevalence of clinics reporting to athenahealth (the provider of our electronic health record data) also vary considerably from state to state, affecting the ability of our models to extract useful information from these data sources. Thus, we examined whether geographical improvement of ARGONet over the benchmark AR52 (as defined by percent reduction of RMSE) are associated with proxies of Google Trends or athenahealth data quality, namely detectable influenza-related search terms from Google Trends and athenahealth population coverage (Supplementary Figure 2a–d). Indeed, linear regression indicates a moderate association of ARGONet improvement with athenahealth coverage and a weak association with detectable influenza-related Google search activity (Supplementary Figure 2e–f). Interestingly,

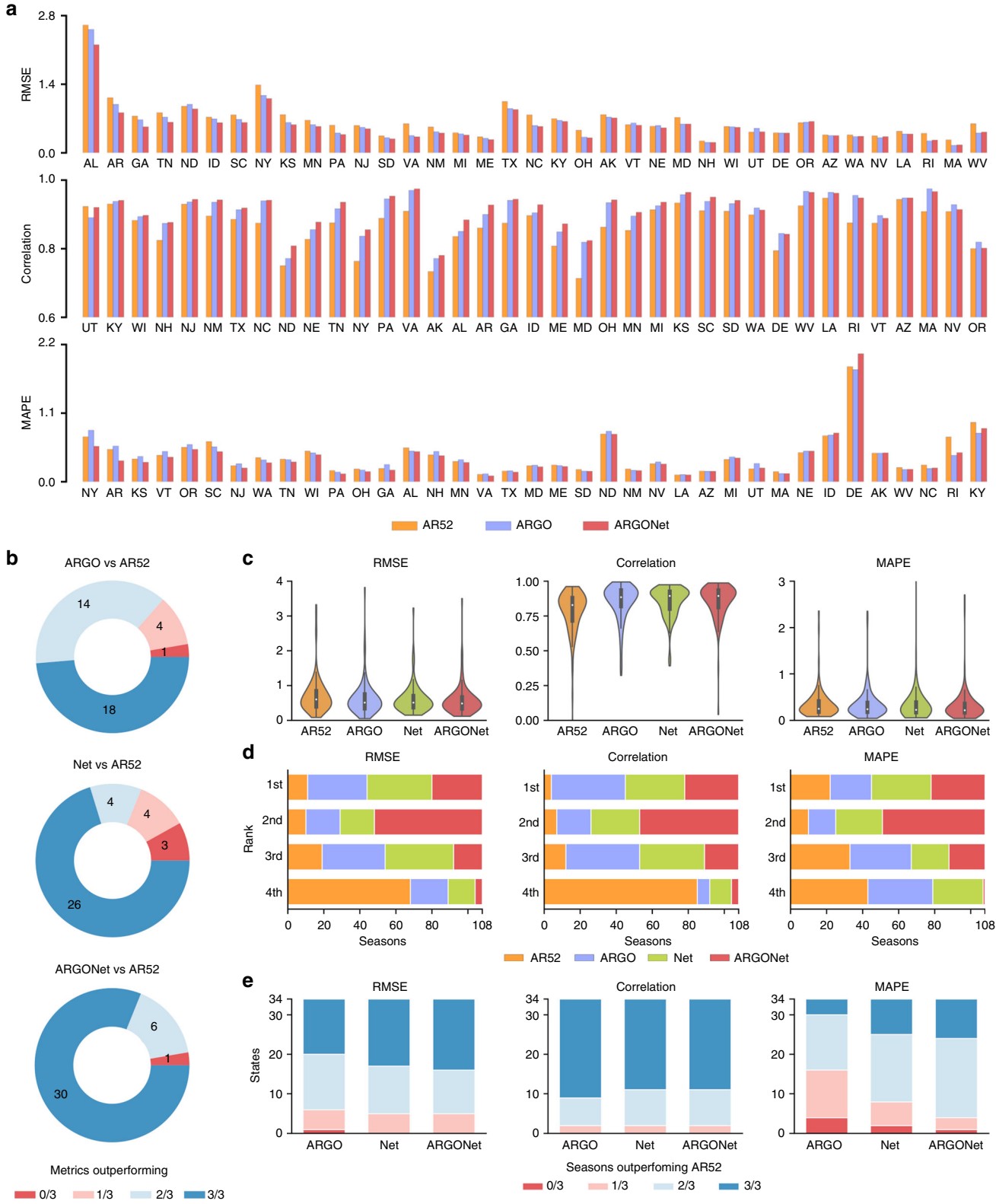

**Fig. 3** Summary of ARGONet performance benchmarked with ARGO and AR52. **a** State-level performance of ARGONet with benchmarks, as measured by RMSE (top), Pearson correlation (middle), and MAPE (bottom), over the period from September 28, 2014 to May 14, 2017. States are ordered by ARGONet performance relative to the benchmarks to facilitate comparison. **b** The proportion of states where ARGO (top), Net (middle), or ARGONet (bottom) outperforms AR52 in 0, 1, 2, or all 3 metrics. **c** The distribution of values for each metric for each model, over the 108 state-seasons during the same period. The embedded boxplots indicate median and interquartile ranges. Numerical values are reported in Supplementary Table 1. **d** The distribution of ranks attained by each model over the 108 state-seasons for each metric. **e** The proportion of states where each model outperforms AR52 in 0, 1, 2, or all 3 influenza seasons within the same period. Only 34 states have data for all three influenza seasons available in this period

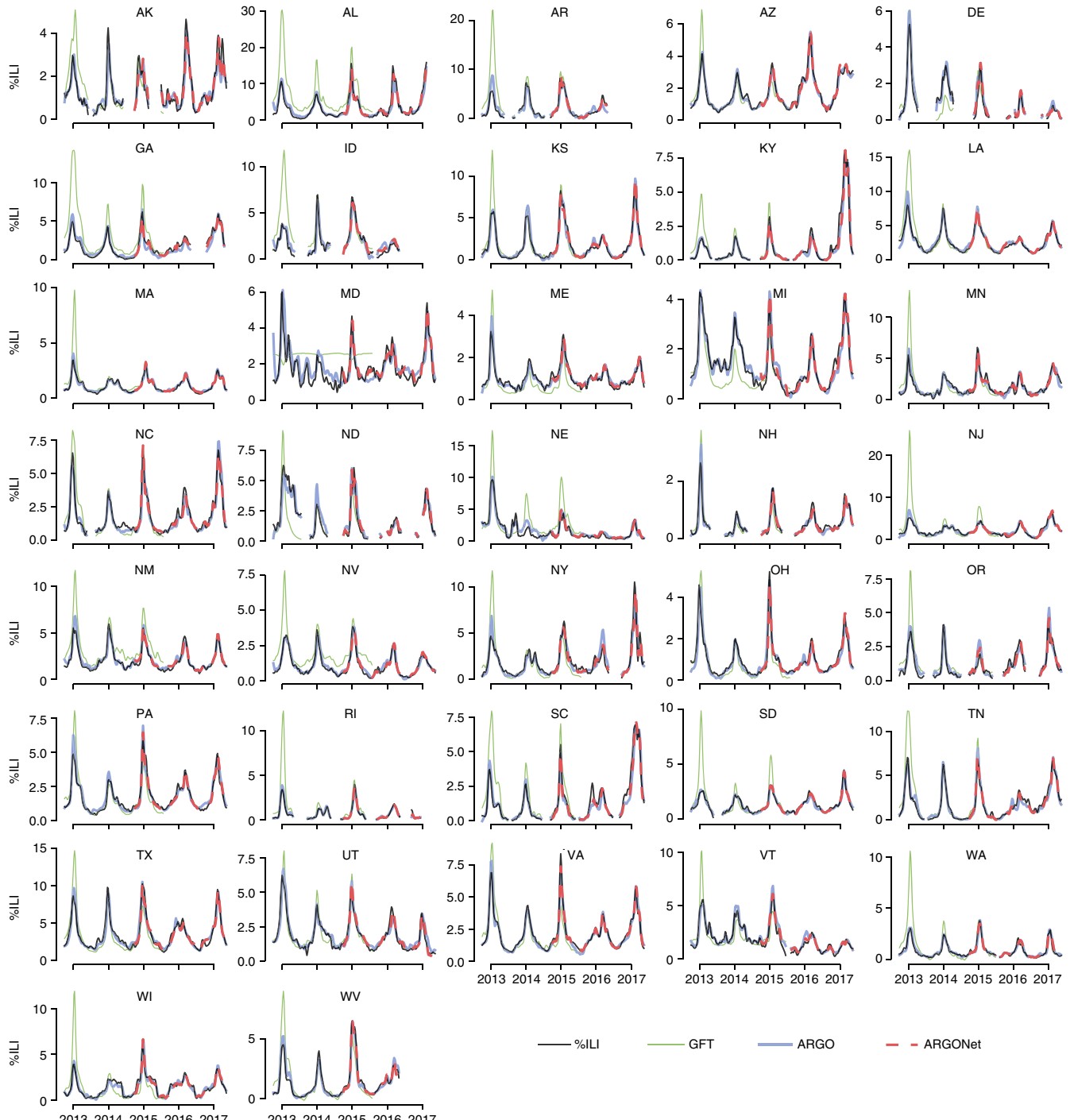

**Fig. 4** Time series plots displaying the performance of ARGO and ARGONet relative to the official CDC %ILI time series. Results are shown over the entire out-of-sample prediction period (September 30, 2012 to May 14, 2017). The GFT benchmark is also shown. Refer to Supplementary Figure 1 for more detailed figures from September 2014–May 2017

influenza-related search activity has a very strong correlation with state population (Supplementary Figure 2g), which suggests that larger pools of Internet users result in better signal-to-noise ratio in search activity. Finally, there is almost no association between ARGONet improvement with the variance of the state's time series and a weak association with the number of healthcare providers reporting to the CDC from each state (Supplementary Figure 2h–i). Future analysis can examine the interplay of these factors with CDC %ILI report quality and structural spatial

correlations. For example, we hypothesize that the Net model contributes strongly in states with lesser-quality data which are adjacent to states with high-quality data. Pairwise %ILI correlation (Fig. 2b) suggests that geographic proximity is a relevant synchronous factor, as many Southeastern states, Western states, and New England/mid-Atlantic states were clustered together.

The lack of unrevised ILI data at the state level over the influenza seasons covered in this study may present a caveat to our results. In real time, ILI estimates reported by the CDC can be

modified, due to post hoc revisions, up to weeks later, complicating model training and testing. While studies modeling influenza have demonstrated important sensitivity analyses in settings with available unrevised data[6,27], such as national ILI, we were not able to analyze this effect. Another pertinent detail concerns the tradeoffs between modeling ILI vs true influenza cases as the outcome variable. We chose ILI as our target because it directly reflects symptomatic patient levels, allowing for rapid population-level interventions such as resource deployment. By nature, our Internet-based data sources (internet searches and outpatient visits) are good proxies of ILI. On the other hand, ILI is not specific only to influenza. Thus, our estimates may reflect a combination of respiratory pathogens, decreasing their utility for mechanistic modeling and virology analysis.

Localized, accurate surveillance of influenza activity can set a foundation for precision public health in infectious diseases. Important developments in this field can involve emerging methodologies for tracking disease at fine-grained spatial resolutions, rapid analysis and response to changing dynamics, and targeted, granular interventions in disparate populations, each of which has the potential to complement traditional public health methods to increase effectiveness of outcomes[28]. We believe that the use of our system can produce valuable real-time subregional information and is a step toward this direction. At the same time, the performance of ARGONet depends directly on the availability and quality of Internet-based input data and also relies on a consistently reporting (even if lagged) healthcare-based surveillance system. We anticipate that data sources will improve over time, for example, if athenahealth continues to gain a larger market share over the states or more Google Trends information becomes available. If these conditions hold, or as more web-based data sources or other electronic health record systems become available in real time, the accuracy of our methods may continue to increase.

## Methods
**Data acquisition**. Three data sources were used in our models: influenza-like illness rates from ILINet, Internet search frequencies from Google Trends, and electronic health records from athenahealth, as described below. Weekly information from each data source was collected for the time period of October 4, 2009 to May 14, 2017.

**Influenza-like illness rates**. Weekly influenza-like illness rates reported by outpatient clinics and health providers for each available state were used as the epidemiological target variable of this study. The weekly rate, denoted %ILI, is computed as the number of visits for influenza-like illness divided by the total number of visits. Data from October 4, 2009 to May 14, 2017 for 37 states were obtained from the CDC. For inclusion in the study, a state must have data from October 2009 to May 2016, with no influenza seasons (week 40 of one year to week 20 of the next) missing. Some states were missing data, usually due to not reporting in the off-season (between week 20 and week 40 of each year). Missing or unreported weeks, as well as weeks where 0 cases were reported, were excluded from analysis on a state-by-state basis. While in real-time ILI values for a given week may be revised in subsequent weeks, we only had access to the revised version of these historical values.

**Internet search frequencies**. Search volumes for specific queries in each state were downloaded through Google Trends, which returns values in the form of frequencies scaled by an unknown constant. While our pipeline used the Google Trends API for efficiency, search volumes can be publicly obtained from www.trends.google.com for reproducibility. Relevant search terms were identified by downloading a complete set of 287 influenza-related search queries for each state, and keeping the terms that were not completely sparse. Because Google Trends left-censors data below an unknown threshold, replacing values with 0, a query with high sparsity indicates low frequency of searches for that query within the state.

In an ideal situation, relevant search queries at the state-level resolution would be obtained by passing the historical %ILI time series for each state into Google Correlate, which returns the most highly correlated search frequencies to an input time series. However, such functionality is only supported at the national level, at least in the publicly accessible tool. Given this limitation, we used two strategies to select search terms:

An initial set of 128 search terms was taken from previous studies tracking influenza at the US national level[6].

To search for additional terms, we submitted multiple state %ILI time series into the Google Correlate and extracted influenza-related terms, under the assumption that some of the state-level terms would show up at the national level.

To minimize overfitting on recent information, the %ILI time series inputted into Google Correlate were restricted from 2009–2013. State-level search frequencies for the union of these terms and the 128 previous terms were then downloaded from the Google Trends API, resulting in 282 terms in total (Supplementary Table 3).

**Electronic health records**. Athenahealth is a cloud-based provider of electronic health records, medical billing, and patient engagement services. Its electronic health records system is currently used by over 100,000 providers across all 50 states. Influenza rates for patients visiting primary care providers over a variety of settings, both inpatient and outpatient, are provided weekly from athenahealth on each Monday. Three types of syndromic reports were used as variables: 'influenza visit counts', 'ILI visit counts', and 'unspecified viral or ILI visit counts', which were converted into percentages by dividing by the total patient visit counts for each week. The athenahealth network and influenza rate variables are detailed in Santillana et al.[24].

**Google flu trends**. In addition to the above data sources, we downloaded GFT estimates as a benchmark for our models. Google Flu Trends provided a public influenza prediction system for each state until its discontinuation in August 2015[29]. GFT values were downloaded and scaled using the same initial training period of 104 weeks used in all of our models (October 4, 2009 to September 23, 2012).

**ARGO model**. The time series prediction framework ARGO (AutoRegression with General Online information) issues influenza predictions by fitting a multivariable linear regression each week on the most recent available Internet predictors and the previous 52 %ILI values. Because of many potentially redundant variables, L1 regularization (Lasso) was applied to produce a parsimonious model by setting the coefficients for weak predictors to 0. The model was re-trained each week on a shifting 104-week training window in order to adapt to the most recent 2 years of data, and the regularization hyperparameter was selected using 10-fold cross-validation on each training set. Details about the ARGO model and its applicability in monitoring infectious diseases such as influenza, dengue, and zika are presented in previous work[5,25,26]. Refer to the Supplement for a detailed mathematical formulation of ARGO.

To fine-tune predictive performance, adjustments to the procedure were introduced on a state-by-state basis:

Filtering features by correlation: For each week, non-autoregressive features ranked outside the top 10 by correlation were removed to reduce noise from poor predictors. Based on previous research, this complementary feature selection process benefits the performance of Lasso, which can be unstable in variable selection[23,25].

Regularization hyperparameters: Features with high correlation to the target variable over the training set received a lower regularization weight, which makes them less subject to the L1 penalty (see the Supplement for derivation).

Weighting recent observations: Although ARGO dynamically trains on the last 104 weeks of observations, more recent observations likely contain more relevant information. Thus the most recent 4 weeks of data received a higher weight (set to be twice the weight of the other variables) in the training set.

**Network-based approach**. Historical CDC %ILI observations show synchronous relationships between states, as shown in Fig. 2a (generated using corrplot[30] with the complete-linkage clustering method). To identify these relationships with the goal of improving our %ILI predictions, for each state, we dynamically constructed a regularized multi-linear model for each week that has the following predictors: % ILI terms for the previous 52 weeks for the target state, and the synchronous (same week's values) and the past three weeks of observed CDC's %ILI terms from each of the 36 other states. Notice that to produce predictions of %ILI for a given state in a given week, the model requires synchronous %ILI from the other states, which would not be available in real-time. Instead, we use ARGO predictions for the current week as surrogates for these unobserved values. Like ARGO, this model is trained with a rolling 104-week window with 10-fold cross-validation to determine the L1 regularizer (formulation in Supplement). This model is denoted Net.

**Ensemble approach**. In order to optimally combine the predictive power of ARGO and Net, we trained an ensemble approach based on a winner-takes-all voting system, which we named ARGONet. ARGONet's prediction for a given week is assigned to be Net's prediction if Net produced lower root mean square error (defined in Comparative analyses) relative to the observed CDC %ILI over the previous K predictions than ARGO. Otherwise, ARGONet's prediction is assigned to be ARGO's prediction. To determine the hyperparameter K for each state, we trained ARGONet using the first 104 out-of-sample predictions of ARGO. We constrained K to take value in {1, 2, 3}. For each state, we utilized and fixed the

identified best value of K to produce out-of-sample predictions for the time period of September 28, 2014 onward. The value of K chosen for each state is shown in Supplementary Table 2.

**Comparative analyses**. To assess the predictive performance of the models, we produced state-level retrospective estimates using two benchmarks: (a) AR52, an autoregressive model, which uses the %ILI from the previous 52 weeks in a Lasso regression to predict %ILI of the current week, and (b) GFT, made by scaling each state's Google Flu Trends time series to its official revised %ILI from October 4, 2009 to September 23, 2012.

The performances of all models and benchmarks compared to the official (revised) %ILI were scored using three metrics: root mean squared error (RMSE), Pearson correlation coefficient, and mean absolute percent error (MAPE). These were computed over the entire study period (September 30, 2012 to May 14, 2017) and over each influenza season (defined as week 40 of one year to week 20 of the next) within the study period.

The models and benchmarks were further scored over two specific sub-periods: (1) the window when GFT was available (September 30, 2012 to August 9, 2015), and (2) the window starting with the first available ARGONet prediction (September 28, 2014 to May 14, 2017).

ARGO model estimates were generated using scikit-learn in Python 2.7[31], while Net and ensemble models were generated in R 3.4.1. Data analysis was conducted in Python except for Fig. 2a, b, which used the R packages corrplot[30] and ggplot[32]. The United States maps in Fig. 2b and Supplementary Figure 2 were made in ggplot using the shapefile from the maps package[33].

**Disclaimer**. The findings and conclusions in this report are those of the authors and do not necessarily represent the official position of the Centers for Disease Control and Prevention.

**Code availability**. The code supporting the results of this study is available from: https://github.com/fl16180/argonet.

## Data availability
The data used in this study are available from Harvard dataverse: [https://doi.org/10.7910/DVN/L5NT70][34] Up-to-date CDC %ILI data can be obtained from CDC's FluView Interactive application: [https://www.cdc.gov/flu/weekly/fluviewinteractive.htm].

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

## Acknowledgements
This work was partially funded by the Centers for Disease Control and Prevention's Cooperative Agreement PPHF 11797-998G-15. The authors thank Josh Gray, Anna Zink, and Dorrie Raymond for the collection and processing of the data from athenahealth. Research reported in this publication was also partially supported by the National Institute of General Medical Sciences of the National Institutes of Health under Award Number R01GM130668. The content is solely the responsibility of the authors and does not necessarily represent the official views of the National Institutes of Health.

## Author contributions
F.L., M.B., and M.S. conceived the study. F.L. and C.L.C. prepared the data for analysis. F.L., M.H., and M.S. designed the computational framework. F.L., M.H., and C.L.C. carried out the implementation of the methods and performed the calculations. F.L., M.H., C.L.C., and M.S. analyzed the data. F.L. and M.S. wrote the first draft of the manuscript. All authors contributed to and approved the final version of the manuscript. M.S. was in charge of the overall direction and planning of the study.

**Additional information**

**Competing interests:** The authors declare no competing interests.

