## [Peer Review File · Nature Communications]

Reviewers' comments:

Reviewer #1 (Remarks to the Author):

This is a well-written paper that describes a machine learning approach to predict flu-like-illness (ILI) in 37 US states. There has been tremendous interest in developing influenza forecasting systems in the past 5-10 years, sparked in part by the annual CDC flu forecasting challenge. This study nicely builds on the existing literature.

I have relatively minor comments:

1) I found the state-clustering analysis presented in Fig 2 interesting, but there is no detail on the approach. What is the method used? What defines these 5 clusters –it does not seem exactly intuitive based on the heatmap in 2a? Are the clusters stable across seasons? Why is MD defined as a singleton? Also the colors in Fig 2b are not defined – does grey represent lack of data? Would it make sense to consider ILI lags between states and revisit 2a?

2) Can the authors elaborate on why the network approach may fare poorly in some states? Also, does the network regression approach considers ILI in the states belonging to the same cluster as the state to be predicted as covariates?

3) It would be useful to add a time series plot showing when ARGO's predictions are selected over NET's predictions. There may be systematic patterns pointing at increased performance of a specific algorithm at the beginning of the epidemic, near the peak, or in other periods of less interest.

4) Fig 3a displays state-level performances, which vary substantially. Do performances scale with the amount of noise in the data or number of physicians reporting to the system?

5) Line 190: why do these 4 states (AR, GA, NY, VT) demonstrate that ARGOnet outperforms ARGO more clearly? Is it because flu dynamics in these states is more closely related to that in other states of the same cluster, or because the local ILI data is so poor in these states that information has to be borrowed from elsewhere?

6) Discussion could address the potential differences between ILI and true influenza virus activity, as ILI may capture the sum of respiratory pathogens and may be less amenable to mechanistic modeling.

7) I was curious about the 52/104-week window in ARGO. Was this selected based on a statistical optimum? Does it reflect memory in the system beyond seasonal effects, eg, residual immunity from the prior season(s)?

Reviewer #2 (Remarks to the Author):

In this study the authors present an machine learning approach to constructing ensemble epidemic forecasts from a suite of individual forecasts. They apply this method to predicting influenza activity for individual states in the United States of America, and demonstrate that the ensemble consistently performs better than any of the individual forecasting methods. There is a growing body of literature which demonstrates that ensemble forecasts are more reliable than individual forecasts for infectious disease epidemics, and the development of improved ensemble methods is therefore timely and valuable. Likewise, the consideration of smaller spatial scales (such as individual states) represents an important progression towards providing forecasts at a scale that is commensurate to that at which public health decisions are made.

Comments

1. Abstract: "The proposed ensemble considerably outperforms each component method in addition to any previously proposed state-specific method for influenza tracking, with higher correlations and lower prediction errors."

The authors have cited several influenza forecasting papers by Jeffrey Shaman's group (references 8, 10, 18). But given the focus on state-level (and smaller) spatial scales in this manuscript, the authors may also wish to consider the following studies, which have considered similar spatial scales:

* Dukic V, et al. Tracking Epidemics With Google Flu Trends Data and a State-Space SEIR Model. *J Am Stat Assoc.* 107, 1410-1426 (2012).

* Yang W, et al. Forecasting Influenza Epidemics in Hong Kong. *PLOS Comput Biol.* 11, e1004383 (2015).

* Yang W, et al. Forecasting Influenza Outbreaks in Boroughs and Neighborhoods of New York City. *PLOS Comput Biol.* 12, e1005201 (2016).

* Moss R, et al. Retrospective forecasting of the 2010--14 Melbourne influenza seasons using multiple surveillance systems. *Epidemiol Infect.* 142, 156-169 (2017).

The Dukic et al. study produced influenza forecasts for 9 separate states in the United States of America, so the authors should certainly examine this study in order to validate the performance claim articulated in the abstract.

This is also relevant to the discussion section, where the authors write "At the state level, where it had not been applied before, it clearly outperformed existing benchmarks over the study period, namely Google Flu Trends and an autoregression."

2. Introduction: "promise to revolutionize the way we treat individual patients and communities in the presence of health threats."

It's not immediately clear to me how using these data streams to estimate current and/or future disease activity will affect the way in which individual patients are treated. In contrast, the utility of such methods to deliver appropriate community-level interventions is apparent.

3. Introduction: "While these measurements are an established indicator of historical ILI activity, they are frequently revised and require around a week to collect from health-care providers across the country, analyze, and report. This delay and potential subsequent revisions can reduce the utility of the system for real-time situational awareness and data analysis."

Later on, the authors state: "Then we generate retrospective influenza estimates using ARGONet ..."

So this study doesn't address the revisions that were made to the original data. They note in the methods section ("Influenza-like illness rates") that they only had access to the revised values of the data (which is certainly a valid reason!). Is it still possible that the original data could be obtained, since they have been made available to each of the teams participating in the CDC's annual FluSight forecasting competition?

This is a real pity, because the subsequent revision of surveillance data is a very real challenge, as

the authors have highlighted in the introduction. To date, I am only aware of one study that has explicitly address this:

* Moss et al. Epidemic forecasts as a tool for public health: interpretation and (re)calibration. *Aust N Z J Public Health*. 42, 69-76 (2018).

4. Introduction: "As a result, these successes at nowcasting in national and regional spatial resolutions are likely not enough to aid decision-making at smaller geographic scales, since important information about local conditions is lost in regional or national aggregates."

And in the discussion: "Future work should explore adding similar approaches to influenza nowcasting systems at finer spatial resolutions, such as the city level."

Certainly there are much fewer city-level forecasting studies than there are at, e.g., national levels. But several do exist in the literature (some of which I have mentioned above). It would be very interesting to see how ensemble forecasts perform at the city level, especially as the number and variety of city-level forecasting methods grows, because it is at these smaller spatial scales that decision-making can really be tailored to the specific circumstances of each region.

Of course, there is also the challenge of smaller case numbers and lower signal-to-noise ratios at smaller spatial scales. The most appropriate scale is almost certain to be (a) very hard to define; and (b) highly specific to the individual setting.

5. Methods, Models: "The model was re-trained each week on a shifting 104-week training window in order to adapt to the most recent two years of data, and the regularization hyperparameter was selected using 10-fold cross validation on each training set."

Does forecast skill decrease when the current season is markedly different from the previous two seasons (e.g., replacement of one A subtype with another)?

6. Methods, Ensemble approach: "To determine the hyperparameter K for each state, we trained ARGONet using the first 104 out-of-sample predictions of ARGO. We constrained K to take value in {1, 2, 3}. The value of K that yielded the lowest RMSE between ARGONet and CDC's %ILI over the training period was chosen to produce out-of-sample predictions in the unseen time period."

What value of K was ultimately chosen for each state? Or were values selected independently at each week and, if so, how did these values vary over time? And how would then apply this method in real-time forecasting?

7. I'm glad to see this study use mean absolute percent error (MAPE), as well as RMSE, because of the substantial variation of scale in case counts/rates during an epidemic. Good choice!

Point by point response to reviewers:

Reviewers' comments:

Reviewer #1 (Remarks to the Author):

This is a well-written paper that describes a machine learning approach to predict flu-like-illness (ILI) in 37 US states. There has been tremendous interest in developing influenza forecasting systems in the past 5-10 years, sparked in part by the annual CDC flu forecasting challenge. This study nicely builds on the existing literature.

I have relatively minor comments:

1) I found the state-clustering analysis presented in Fig 2 interesting, but there is no detail on the approach. What is the method used?

We used the complete-linkage method which was the default setting on the corrploth package in R. We have added this to the methods:

“Historical CDC %ILI observations show synchronous relationships between states, as shown in Fig. 2a (generated using corrploth [31] with the complete-linkage clustering method).”

What defines these 5 clusters –it does not seem exactly intuitive based on the heatmap in 2a?

The states are hierarchically divided into 5 groups using pairwise correlations as the similarity measure. It turned out that mathematically these 5 clusters of states are more similar within their own group than to the outside groups. The heatmap alone shows pairwise information but not similarities among >2 states.

Are the clusters stable across seasons?

This is an interesting question that should be investigated in a future study. We chose not to focus on this descriptive aspect because our proposed data-driven network methodology does not necessarily select the states within these clusters as predictors. Instead, our model automatically chooses which states to include as predictors based on a data-driven regularized approach (LASSO) that also removes redundant information based on the available historical data available at the time of prediction.

On the other hand, we chose to include a visualization of these clusters as a way to communicate the motivation behind our use of neighboring flu activity for prediction purposes. A future study could be conducted to analyze these clusters and their stability from an inferential perspective.

Why is MD defined as a singleton?

MD is a singleton because its ILI curve is more distant from any other state's ILI curve than any of the other states are to each other. The hierarchical clustering thus first separates MD as its own state group. We believe this to reflect unusual (possibly unreliable) characteristics of ILI reporting in MD, as our ARGO/ARGONet methods also perform far worse in this state than any other.

Also the colors in Fig 2b are not defined – does grey represent lack of data?

Thank you for pointing out the ambiguity in Fig 2b. We have updated the figure with a legend to indicate that grey represents states without data.

Would it make sense to consider ILI lags between states and revisit 2a?

Fig. 2a is a diagnostic for which we focused on concurrent ILI without lags for simplicity. While it would be interesting to include ILI lags in determining similarities for Figure 2a, deciding how to incorporate them would be complicated and may make the figure more difficult to interpret.

In addition, this lagged information is taken into account by our approach. In fact, our Net methodology can choose ILI lags 1-3 (in addition to concurrent ILI) from all states as input variables for prediction. Finally, what is displayed on this figure does not directly explain the quality and/or performance of our methods.

It would certainly make sense, however, for a future study focusing on analyzing statistical relationships between these states to address your question more thoroughly.

2) Can the authors elaborate on why the network approach may fare poorly in some states?

Thank you for this suggestion. We have some ideas around this and we have added the following paragraph to the Discussion:

“A large factor determining the success of our approach is whether covariates that have a strong association with the response variable, over the training set (up to 2 years prior), maintain their behavior in the present. This may be especially important in the network approach, because the selected predictors capture underlying historical geographical relationships. The ILI between two neighboring states may be highly correlated over some seasons, but changes in influenza dynamics over the next season may induce their ILI trends to behave differently. In such cases, the relationship apparent in the older data would provide an inaccurate picture of the present.”

That being said, while Net may perform worse than ARGO in some states, it still fares well in general. We have revised our Supplementary Figure 1 (following your suggestion in the next question) to show a plot within each state indicating whether our ensemble model chose ARGO or Net’s prediction each week. It turns out that both ARGO and Net are selected with relatively even proportions.

Also, does the network regression approach considers ILI in the states belonging to the same cluster as the state to be predicted as covariates?

The network regression uses ILI from all the states as potential covariates, not just the ones in the same cluster. Moreover, the covariates selected to contribute to the regression are selected with a data-driven approach by using L1 regularization, and thus may be different each week. This, to some extent, suggests that the clusters may not be stable across seasons. We have clarified the methods to reflect this:

“To identify these relationships with the goal of improving our %ILI predictions, for each state, we dynamically constructed a regularized multi-linear model for each week that has the following predictors: %ILI terms for the previous 52 weeks for the target state, and the synchronous (same week’s values) and the past three week’s of observed CDC’s %ILI terms from each of the 36 other states.”

3) It would be useful to add a time series plot showing when ARGO’s predictions are selected over NET’s predictions. There may be systematic patterns pointing at increased performance of a specific algorithm at the beginning of the epidemic, near the peak, or in other periods of less interest.

Thank you for this excellent idea. We have done so in the revised version of the paper (please see Supplementary Figure 1) and made reference to it in the results. It appears that both ARGO and Net’s predictions are frequently chosen without a clear pattern in time.

4) Fig 3a displays state-level performances, which vary substantially. Do performances scale with the amount of noise in the data or number of physicians reporting to the system?

In this particular study, it is not clear how to quantify noise in the data. The reason behind this is that it is impossible to measure the true number of ILI cases across the entire state. As such, we cannot compare the true ILI values with the reported values, as captured by ILINet. In other words, measurement error is not possible to assess here.

On the other hand, the reviewer may be referring to the values of the variance in the time series itself as noise(?), which can be computed. We decided to use a standard measure of time series noise: $\text{stdev}(y) / \text{mean}(y)$.

While we do not have access to the number of physicians reporting to the system, we do have data on the number of providers at (i.e. institutional/clinical level) reporting in each state.

We have added regressions corresponding to these two questions to the existing regressions in Supplement Figure 2. We added comments in the Discussion as shown here:

“Finally, there is almost no association between ARGONet improvement with the variance of the state’s time series and a weak association with the number of healthcare providers reporting to the CDC from each state (Supplementary Figure 2h-i).”

5) Line 190: why do these 4 states (AR, GA, NY, VT) demonstrate that ARGONet outperforms ARGO more clearly? Is it because flu dynamics in these states is more closely related to that in other states of the same cluster, or because the local ILI data is so poor in these states that information has to be borrowed from elsewhere?

While these questions are interesting and thought-provoking, they cannot be answered conclusively with the tools and data we currently have. For example, we cannot accurately assess whether the local ILI data is poor simply because the ILI is our gold standard for the study. We don’t have a higher quality standard (ground truth) to compare against. Comparing flu dynamics between states would be good to study from a mechanistic/epidemiological perspective in the future, but would likely require more data than the data on ILI alone.

However, we can give some educated guesses for these states. In general, either the input variables in these states for ARGO are not as strong as the input variables for Net, or the ensemble is especially stable in these states. VT and AR are in the first case, as Net itself strongly outperforms ARGO (Fig. 2c), which may have to do with local data such as Google Trends or electronic health records not being strong predictors of ILI. On the other hand, NY and GA seem to be in the second group. Flu dynamics may play a role NY’s ensemble stability because of its population density and the proximity of NYC to multiple neighboring states.

Since we had arbitrarily picked these states based on their performance, we decided to remove the sentence pointing them out to minimize confusion to the reader.

6) Discussion could address the potential differences between ILI and true influenza virus activity, as ILI may capture the sum of respiratory pathogens and may be less amenable to mechanistic modeling.

That is a good point. There are very relevant pros and cons of ILI vs true influenza modeling. The benefit of ILI is that it is potentially more useful for immediate population-level interventions such as scheduling and deploying resources that may not care if the patient really has influenza. On the other hand, as you stated, true influenza is amenable to modeling and is also relevant for other situations such as vaccine development.

We have added these points in the Discussion:

“Another pertinent detail concerns the tradeoffs between modeling ILI vs true influenza cases as the outcome variable. We chose ILI as our target because it directly reflects symptomatic patient levels, allowing for rapid population-level interventions such as resource deployment. By nature, our Internet-based data sources (internet searches and outpatient visits) are good proxies of ILI. On the other hand, ILI is not specific only to influenza. Thus, our estimates may reflect a combination of respiratory pathogens, decreasing their utility for mechanistic modeling and virology analysis.”

7) I was curious about the 52/104-week window in ARGO. Was this selected based on a statistical optimum? Does it reflect memory in the system beyond seasonal effects, eg, residual immunity from the prior season(s)?

Thank you for your helpful questions and comments. The original formulation of the ARGO model (*) developed the 52/104 window as a constrained data-driven decision. The use of 52 autoregressive terms makes sense in terms of seasonality because it is a full annual cycle of data. For the 104-week training period, the important justification is that the relationship between the response variable and the predictors, baseline Google search activity and EHR data, may change over time. If these relationships were stable, then it would make sense to use all available past data. However, the athenahealth network has grown significantly over the years, and search patterns for Google terms change over time. Thus the 2 year window is a compromise to focus on recent conditions while still getting enough data to train on. Empirically, these parameters seem to behave well for us in a variety of situations. On the other hand, we tried to avoid overfitting to data by testing too many different windows.

(*) Yang S, Santillana M, Kou SC. Accurate influenza epidemics estimation via ARGO. Proceedings of the National Academy of Sciences Nov 2015, 112 (47)

Reviewer #2 (Remarks to the Author):

In this study the authors present an machine learning approach to constructing ensemble epidemic forecasts from a suite of individual forecasts. They apply this method to predicting influenza activity for individual states in the United States of America, and demonstrate that the ensemble consistently performs better than any of the individual forecasting methods. There is a growing body of literature which demonstrates that ensemble forecasts are more reliable than individual forecasts for infectious disease epidemics, and the development of improved ensemble methods is therefore timely and valuable. Likewise, the consideration of smaller spatial scales (such as individual states) represents an important progression towards providing forecasts at a scale that is commensurate to that at which public health decisions are made.

Comments

1. Abstract: "The proposed ensemble considerably outperforms each component method in addition to any previously proposed state-specific method for influenza tracking, with higher correlations and lower prediction errors."

The authors have cited several influenza forecasting papers by Jeffrey Shaman's group (references 8, 10, 18). But given the focus on state-level (and smaller) spatial scales in this manuscript, the authors may also wish to consider the following studies, which have considered similar spatial scales:

* Dukic V, et al. Tracking Epidemics With Google Flu Trends Data and a State-Space SEIR Model. *J Am Stat Assoc.* 107, 1410-1426 (2012).

* Yang W, et al. Forecasting Influenza Epidemics in Hong Kong. *PLOS Comput Biol.* 11, e1004383 (2015).

* Yang W, et al. Forecasting Influenza Outbreaks in Boroughs and Neighborhoods of New York City. *PLOS Comput Biol.* 12, e1005201 (2016).

* Moss R, et al. Retrospective forecasting of the 2010--14 Melbourne influenza seasons using multiple surveillance systems. *Epidemiol Infect.* 142, 156-169 (2017).

The Dukic et al. study produced influenza forecasts for 9 separate states in the United States of America, so the authors should certainly examine this study in order to validate the performance claim articulated in the abstract.

This is also relevant to the discussion section, where the authors write "At the state level, where it had not been applied before, it clearly outperformed existing benchmarks over the study period, namely Google Flu Trends and an autoregression."

Thank you for bringing these interesting papers to our attention. We have added these references in our overview in the introduction section in the following sentences:

“A detailed statistical analysis by Dukic, Lopes, and Polson (2012) presented a Bayesian state-space SEIR model over 9 states but did not attempt out-of-sample prediction [19]. Other studies have demonstrated the feasibility of influenza estimation at smaller spatial scales [20-23], though they have not yet extended their results to multiple locations.”

We also added reference in the Discussion:

“While previous studies have established the feasibility of city-level flu nowcasting [21-23], the results from this study suggest that ensemble methodologies incorporating flu activity from neighboring cities may improve prediction stability and accuracy at such spatial resolutions as well. [...]”

2. Introduction: "promise to revolutionize the way we treat individual patients and communities in the presence of health threats."

It's not immediately clear to me how using these data streams to estimate current and/or future disease activity will affect the way in which individual patients are treated. In contrast, the utility of such methods to deliver appropriate community-level interventions is apparent.

We agree that our statement did not make the most logical connection. We have modified the sentence to: *“promise to revolutionize the identification and delivery of community-level interventions in the presence of health threats.”*

3. Introduction: "While these measurements are an established indicator of historical ILI activity, they are frequently revised and require around a week to collect from health-care providers across the country, analyze, and report. This delay and potential subsequent revisions can reduce the utility of the system for real-time situational awareness and data analysis."

Later on, the authors state: "Then we generate retrospective influenza estimates using ARGONet ..."

So this study doesn't address the revisions that were made to the original data. They note in the methods section ("Influenza-like illness rates") that they only had access to the revised values of the data (which is certainly a valid reason!). Is it still possible that the original data could be obtained, since they have been made available to each of the teams participating in the CDC's annual FluSight forecasting competition?

This is a real pity, because the subsequent revision of surveillance data is a very real challenge, as the authors have highlighted in the introduction. To date, I am only aware of one study that has explicitly address this:

* Moss et al. Epidemic forecasts as a tool for public health: interpretation and (re)calibration. Aust N Z J Public Health. 42, 69-76 (2018).

Thank you for your observation. We agree that this is an important issue. Indeed, the impact of revisions (or the “backfill” effect) on real-time prediction is considerable and thus should be studied.

The ILI historical unrevised data is available from the CDC for the national and regional levels. But at the state level, this data is not available until the 2017-18 season, when the CDC started actively releasing the data for the new FluSight state-level forecasting contest. (One of the co-authors of this study is a part of the CDC team managing the contest and confirmed this.) Thus, in the absence of access of this historical data, we did our best to study the feasibility of our methods.

We agree that the paper you mentioned explicitly addresses the importance of revisions. We have added a section in the Discussion addressing this issue and cited the paper. In case it is of interest, another paper that addressed ILI forecast performance pre- and post-revision is:

Yang, Shihao, Mauricio Santillana, and Samuel C. Kou. "Accurate estimation of influenza epidemics using Google search data via ARGO." Proceedings of the National Academy of Sciences 112.47 (2015): 14473-14478.

In Discussion:

“The lack of unrevised ILI data at the state level over the influenza seasons covered in this study may present a caveat to our results. In real time, ILI estimates reported by the CDC can be modified, due to post-hoc revisions, up to weeks later, complicating model training and testing. While studies modeling influenza have demonstrated important sensitivity analyses in settings with available unrevised data [6,27], such as national ILI, we were not able to analyze this effect.”

4. Introduction: "As a result, these successes at nowcasting in national and regional spatial resolutions are likely not enough to aid decision-making at smaller geographic scales, since important information about local conditions is lost in regional or national aggregates."

And in the discussion: "Future work should explore adding similar approaches to influenza nowcasting systems at finer spatial resolutions, such as the city level."

Certainly there are much fewer city-level forecasting studies than there are at, e.g., national levels. But several do exist in the literature (some of which I have mentioned above). It would be very interesting to see how ensemble forecasts perform at the city level, especially as the number and variety of city-level forecasting methods grows, because it is at these smaller

spatial scales that decision-making can really be tailored to the specific circumstances of each region.

Of course, there is also the challenge of smaller case numbers and lower signal-to-noise ratios at smaller spatial scales. The most appropriate scale is almost certain to be (a) very hard to define; and (b) highly specific to the individual setting.

Thank you for the insightful comments. City-level forecasting has fortunately been picking up traction recently, and there are valuable studies that have started to examine the topic in depth. As you mentioned, poorer signal-to-noise ratio and data availability tends to complicate small-scale prediction. Because of the additional complexities, these models are also more challenging to validate and adapt over a wide scale (i.e. applying similar methodology effectively to multiple cities or states), which may be why existing studies tend to focus only on a single (or a few) cities at a time. Thus, we believe that it would be ideal to develop robust methods that can function at multiple locations without over-intensive modifications. That is why it was important in our study to develop models that simultaneously could work on as many states as possible.

It has also been of interest to us to see how ensemble methods fare at finer spatial resolutions. Some early work has indicated that ensemble methods are advantageous in these data-poorer settings because they increase stability. We have a recent study at the Boston city level that found significant improvements from using various ensemble methods compared to individual models:

Lu, F. S. et al. Accurate Influenza Monitoring and Forecasting Using Novel Internet Data Streams: A Case Study in the Boston Metropolis. *JMIR Public Health Surveill* 4, e4 (2018).

We have qualified our statement in the Discussion in response to these nuances as well as the previous studies at finer spatial resolutions that you mentioned:

"While previous studies have established the feasibility of city-level nowcasting in specific locations [21-23], the results from this study suggest that ensemble methodologies incorporating flu activity from neighboring cities may improve prediction stability and accuracy at such spatial resolutions as well. This would be of particular interest given that finer spatial resolutions experience fewer numbers of flu cases, which in turn makes the process of finding a meaningful signal in Internet-based data sources a bigger challenge."

5. Methods, Models: "The model was re-trained each week on a shifting 104-week training window in order to adapt to the most recent two years of data, and the regularization hyperparameter was selected using 10-fold cross validation on each training set."

Does forecast skill decrease when the current season is markedly different from the previous two seasons (e.g., replacement of one A subtype with another)?

We think this question is important and should be addressed perhaps with a separate more in-depth study that establishes/defines when a flu season is similar (or different) to previous ones, in terms of subtypes of flu (A vs B). Some other studies have taken into account this, such as:

Brooks, Logan C., David C. Farrow, Sangwon Hyun, Ryan J. Tibshirani, and Roni Rosenfeld. "Flexible modeling of epidemics with an empirical Bayes framework." *PLoS computational biology* 11, no. 8 (2015): e1004382.

It is hard to assess with any confidence whether our forecast skill is different based on seasonal differences based on the limited seasonal sample set. However, we should note that our models use two data sources (Google searches and EHR) that are based on queried/reported flu symptoms and thus should be subtype-agnostic as long as the symptoms have not changed between different flu strains across seasons. Therefore, our models should also be robust and subtype-agnostic.

6. Methods, Ensemble approach: "To determine the hyperparameter K for each state, we trained ARGONet using the first 104 out-of-sample predictions of ARGO. We constrained K to take value in {1, 2, 3}. The value of K that yielded the lowest RMSE between ARGONet and CDC's %ILI over the training period was chosen to produce out-of-sample predictions in the unseen time period."

What value of K was ultimately chosen for each state? Or were values selected independently at each week and, if so, how did these values vary over time? And how would then apply this method in real-time forecasting?

We tested the three possible values of K for all the states. The value of K that led to the best performance for each state during the training period was chosen and fixed for the entire holdout time period. We added a table to the Supplement indicating the value of K for each state. We realized our original description in the Methods should be clarified. We have updated it in the revised version of our manuscript to:

"To determine the hyperparameter K for each state, we trained ARGONet using the first 104 out-of-sample predictions of ARGO. We constrained K to take value in {1, 2, 3}. For each state, we utilized and fixed the identified best value of K to produce out-of-sample predictions for the time period of September 28, 2014 onward. The value of K chosen for each state is shown in Supplementary Table 2."

Training this method in real-time would not be an issue since our hyperparameter is trained on 2 years of previously-seen data (which is the reason why ARGONet predictions start 2 years after the other models). In real-time a reasonable strategy may be to retrain the hyperparameters at the start of each flu season using the last two (or more) seasons as training data.

7. I'm glad to see this study use mean absolute percent error (MAPE), as well as RMSE, because of the substantial variation of scale in case counts/rates during an epidemic. Good choice!

We agree. Thank you for the useful comments and suggestions.

REVIEWERS' COMMENTS:

Reviewer #1 (Remarks to the Author):

The authors have fully addressed my comments

Reviewer #2 (Remarks to the Author):

The authors have carefully and thoroughly addressed all of my comments.

And thank you for bringing two papers ("Accurate estimation of influenza epidemics using Google search data via ARGO", and "Accurate Influenza Monitoring and Forecasting Using Novel Internet Data Streams: A Case Study in the Boston Metropolis") to my attention, neither of which I had previously seen.